# A Comparison of Bilateral vs. Unilateral Flywheel Strength Training on Physical Performance in Youth Male Basketball Players

**DOI:** 10.3390/jfmk10010081

**Published:** 2025-02-27

**Authors:** Bogdan Belegišanin, Nikola Andrić, Tatjana Jezdimirović Stojanović, Alen Ninkov, Gordan Bajić, Nedžad Osmankač, Mladen Mikić, Marko D. M. Stojanović

**Affiliations:** 1Faculty of Sport and Physical Education, University of Novi Sad, 21000 Novi Sad, Serbia; belegisaninb@gmail.com (B.B.); nikola.trenaznaekspertiza18@gmail.com (N.A.); mmmikac@gmail.com (M.M.); 2Training Expertise, 21000 Novi Sad, Serbia; tatjanaj.ns@gmail.com (T.J.S.); ninkov7@gmail.com (A.N.); 3Faculty of Health Sciences, Apeiron University, 78000 Banja Luka, Bosnia and Herzegovina; gordan.z.bajic@apeiron-edu.eu; 4The Provincial Institute for Sports and Sports Medicine, 21000 Novi Sad, Serbia; nedja@mail.ru

**Keywords:** unilateral, bilateral, vertical jump, strength, change of direction ability, reactive strength index

## Abstract

**Background/objectives:** This study aimed to compare the effects of bilateral and unilateral flywheel training programs on leg strength, sprint performance, jumping, and change of direction ability in young basketball players. **Methods:** Twenty-two youth male basketball players were randomly assigned to two groups: the unilateral group (UG; n = 11; age = 15.5 ± 0.5 years) and the bilateral group (BG; n = 11; age = 15.2 ± 0.4 years). Both groups participated in a six-week flywheel training intervention (UG: split squat; BG: half squat) alongside their regular basketball activities. Performance measures included change of direction ability (5-0-5 test), knee extension 60 degrees/s leg strength (EX60), bilateral and unilateral countermovement jump heights (CMJ, CMJL, and CMJD), reactive strength index (RSI), and 5 m and 20 m sprint times (SPR5m and SPR20m). A 2 × 2 ANOVA was used to evaluate pre- to post-intervention changes. **Results:** Significant interaction effects were observed for the 5-0-5 test (F = 13.27; *p* = 0.02), with pre–post improvements of 8.4% and 13.3% for the BG and UG, respectively. Both groups showed significant CMJ improvements (11.4%, ES = 0.69 for the BG; 14.6%, ES = 1.4 for the UG). The UG demonstrated greater unilateral jump improvements compared to the BG. Significant RSI improvements were found for both groups (BG: 19.6%, ES = 0.95; UG: 19.6%, ES = 0.77). Both groups improved on sprint performance, with the UG showing slightly larger effect sizes. **Conclusions:** Unilateral flywheel strength training appeared to be a more effective strategy than bilateral training for enhancing strength, sprinting, jumping, and change of direction ability in youth basketball players.

## 1. Introduction

Basketball is a team sport played on a court that involves alternating bursts of high-intensity activities lasting various lengths of time, separated by periods of low to moderate intensity [1]. Basketball players need to perform a range of high-intensity actions during the game, including accelerations, jumps, sprints, and direction changes, and sport-specific skills like shuffling, pivoting, and dribble penetration [2]. It has been shown that the quality and quantity of these activities are strong determinants of both team and player success [3,4]. In addition, research shows that resistance training positively influences the performance of these activities [5]. So, to help players improve on game performance, resistance training is considered an essential part of players’ training schedules.

One of the challenges practitioners face is identifying the exercises that will optimize the training effects of resistance programs. Typically, resistance exercises are categorized as either unilateral or bilateral. Unilateral exercises involve a weight-bearing movement that primarily engages one limb, such as a single-leg squat or Bulgarian split squat. In contrast, bilateral exercises involve both limbs performing equally and simultaneously, such as in a back squat or deadlift. Traditionally, bilateral exercises have been prioritized for athletic development [6] because of their proven benefits in enhancing strength and power [7,8]. Improvements in bilateral strength and power have been associated with enhancements in related movements, such as change of direction speed [9], sprint [10], and jump height [11].

However, many essential skills specific to basketball (e.g., running, side-stepping, changing direction, and lay-up) are executed unilaterally. Based on the dynamic correspondence principle [12] and the tendency to maximize training transfer [13,14], unilateral resistance training has been prioritized lately in performance-oriented resistance training programs [15]. Research has demonstrated the advantages of unilateral training in enhancing the change of direction speed, jumping performance, and sprint times [16,17,18,19]. Given that both bilateral and unilateral training have been found to boost different aspects of athletic performance, it would be beneficial to determine if one training method is more effective than the other. This remains unclear, with equivocal findings in two recently published meta-analyses [20,21].

Moreover, the occurrence of eccentric movements in basketball—such as jumps, decelerations, changes in direction, and sprints [22]—has sparked growing interest in incorporating eccentric-focused exercises into basketball training programs to enhance performance and overall athletic conditioning [23]. Research indicates that resistance training regimens that effectively target the eccentric phase of muscle contraction may lead to greater neuromuscular adaptations than conventional resistance training methods [24,25]. Different training modalities have been designed to accentuate the eccentric phase of movement [26], with the use of the inertia of a rotating flywheel, known as flywheel training, being the most frequently used and also very effective for improving strength, countermovement jump (CMJ) performance, sprint time, and change of direction ability (COD) in both the general athletic population [27] and basketball players [28].

Interestingly, there are few studies that have investigated the effects of flywheel unilateral vs. bilateral resistance training on athletic performance in athletes generally [29,30], with only one study conducted on a sample of basketball players [31]. As a result, the impact of bilateral versus unilateral flywheel strength training on athletic performance in this team sport population remains ambiguous. Consequently, this study aimed to evaluate the long-term effects of unilateral (lateral lunge) versus bilateral (half squat) flywheel training on strength, jumping ability, speed, and agility in youth basketball players. We hypothesized that unilateral flywheel training would lead to greater improvements in fitness attributes.

## 2. Materials and Methods

### 2.1. Participants

A total of twenty-four youth basketball players volunteered to participate in the study and were randomly allocated to either the unilateral flywheel training (UG, n = 11; age = 15.5 ± 0.5 years; stature = 186 ± 6 cm; body mass = 71 ± 12 kg) or bilateral flywheel training (BG, n = 11; age = 15.2 ± 0.4 years; stature = 183 ± 4 cm; body mass = 69 ± 7 kg) groups through a simple randomization process that utilized a random number generator.

Participants in the study were from two different basketball clubs in Novi Sad, located in the province of Vojvodina, Serbia. Both clubs were active in the regional league, one level below the national league, throughout the intervention period.

Inclusion criteria mandated that participants had (1) no injuries or illnesses in the four months prior to the intervention and (2) at least four years of experience in training. The players attended basketball training sessions four times a week, with each session lasting around 90 min, in addition to participating in one competitive match weekly. In addition, all participants had at least two years of experience in resistance training gained from their time in previous age group teams, although none of them had any prior experience with flywheel training.

Before engaging in the study, all participants were informed about the risks, benefits, and objectives of the research. While no injuries were reported during the training interventions, two players had to withdraw due to medical issues unrelated to mechanical injuries. Those who completed the study maintained over 90% strength training session attendance. Throughout the intervention, none of the players took medications or dietary supplements.

The study followed the Declaration of Helsinki (2008) regarding medical research involving human participants, and the ethics committee of the University of Novi Sad in Serbia reviewed and approved the study protocol (Ref. No. 33-02-07/2023-6, approval date 16 January 2024). All participants were provided with both verbal and written information about the study, and written consent for participation was voluntarily given by their parents or legal guardians.

### 2.2. Study Design

A longitudinal experimental study employing a between-subjects design was conducted to evaluate the effects of unilateral versus bilateral flywheel training within a 6-week resistance training regimen. The training interventions were implemented during the latter phase of the competitive season, specifically from April to May 2024. Baseline assessments were performed a week prior to the onset of the first training session. Following the initial assessment, participants in both groups engaged in 3 to 4 sets of 5 to 6 repetitions of their assigned exercises—split squats for the unilateral group (UG) and half squats for the bilateral group (BG)—to acclimate to the training protocol. Additionally, both groups participated in a supplementary familiarization session four days before the commencement of the program. After the warm-up, the group of players was evenly divided into three stations, with one conditioning coach assigned to each station to implement the training program. Each station had pre-prepared lists indicating group assignments, where each completed set, the number of repetitions, and any pertinent comments were recorded. Circuit training methodology was employed, with all participants at a given station completing the set before proceeding to the next one.

Both initial and final testing were conducted across two sessions. A uniform warm-up routine comprising light jogging, dynamic stretching, and lower limb activation exercises was conducted before each test. On the first day, the participants’ anthropometric characteristics were measured before conducting the 20 m sprint (with 5 m split times) and change of direction (COD) tests on the basketball court. Two or three days later, the subjects completed jumping tests and strength assessments in the specialized performance lab.

Evening supervised strength training sessions for the groups took place at the basketball sports center, utilizing three isoinertial devices (kBox; Exxentric AB, Bromma, Sweden) to ensure a seamless training experience. Each session was conducted by three knowledgeable strength and conditioning coaches with substantial expertise in flywheel training, guaranteeing participants received top-notch instruction. A uniform warm-up was conducted before each strength training, which included 3 min of jogging, 5 min of mobility exercises, 10 bodyweight squats (BG), and 5 linear and lateral lunges on each leg (UG).

The final testing took place five days after the training intervention, utilizing the same tests, in the same sequence, at the same time of day, and by the same examiners as during the baseline assessment. Participants were advised to avoid any strenuous activities for 48 h leading up to the final testing.

### 2.3. Measurements

#### 2.3.1. Anthropometrics

Anthropometric measurements were conducted by a Level 3 anthropometrist certified by the International Society for Advancement in Kinanthropometry (ISAK), adhering to standardized protocols. The technical error of measurement (TEM) for height and body mass was under 0.02%. Height was measured with a SECA rod (Seca GmbH, Hamburg, Germany), which has a precision of 1 mm and a range of 130–210 cm. Body mass was assessed using a SECA scale with a precision of 0.1 kg and a range of 2–130 kg.

#### 2.3.2. CMJ (Countermovement Jump) Test

The countermovement jump test is a validated, reliable, and commonly used test for assessing leg power [32]. The CMJ test was performed on a portable force plate (K-Deltas, Kinvent Inc., Montpellier, France), sampling at 1000 Hz. Subjects were instructed to keep their hands akimbo during the test. Starting at standing position, participants swiftly went downward to a self-selected semi-squat position, and then accelerated and jumped upward maximally, keeping hands on hips. Three trials were performed, with 45 s of passive recovery. The best jump performance was selected for further analysis.

#### 2.3.3. SL CMJ R/L (Single-Leg Countermovement Jump Right/Left Leg) Test

SL CMJ R/L tests were measured using the portable force plate system (K-Deltas, Kinvent Inc., Montpellier, France). Subjects were instructed to keep their hands on their hips during the test and not to swing with the opposite leg. The test started in an upright position with the right/left leg on the ground and the left/right leg in the air flexed at 90 degrees in the knee joint. Subjects were instructed to brake downward to a self-selected single-leg semi-squat position, then accelerate with hands on the hips and jump up maximally in the air, landing on the jumping leg with the knee extended. Three trials with 45 s of rest were performed, with the best result used for further statistical analysis.

#### 2.3.4. DJ–RSI (Drop Jump–Reactive Strength Index) Test

The reactive strength index is an established metric for evaluating an individual’s reactive strength and their capacity to withstand significant eccentric forces [33]. In the DJ test, participants stepped off a 40 cm box, landed on a force plate bilaterally with hands on the hips, and jumped maximally. They were instructed to perform jumps as high as possible while spending the shortest time possible on the force plate. Each participant performed the task three times, with a 30 s rest period, and the peak jump height was recorded for subsequent analysis. The reactive strength index was calculated by dividing the flight time by the contact time (in seconds), with the best result utilized for further analysis.

#### 2.3.5. 5 m and 20 m Sprint Tests

Sprint tests, which measure the time required to cover a specified distance (commonly referred to as split times), have demonstrated valid and reliable measurements of athletes’ speed [34].

Participants completed a 20 m sprint test with 5 m split times, recorded using photocells (Microgate—Witty, Bolzano, Italy). Following a specific warm-up, they performed two sub-maximal efforts (approximately 90% of their maximum speed), after which they undertook two sprint trials with a 2 min rest. Subjects began from a split stance, 0.3 m from the light gate. Participants were instructed to sprint through the poles at their maximum speed whenever they chose to, following the signal “go”. During the test, participants were highly encouraged to give their best effort. The top results were utilized for statistical analysis.

#### 2.3.6. COD 5-0-5 (Change of Direction 5-0-5) Test

The 5-0-5 test is typically used to measure the change of direction ability across various athlete populations [35]. In the 5-0-5 COD test, two timing gates were positioned 5 m from a marked turning point. Players began their run 10 m from the timing gates (Microgate—Witty Italy), which placed them 15 m from the turning point. They were instructed to accelerate as quickly as possible through the timing gates, then decelerate at the 15 m mark before sprinting back through the timing gates as swiftly as possible. Each player performed two trials with turns off both left and right legs, and the best result was used for analysis.

#### 2.3.7. Isokinetic Strength Tests

Isokinetic dynamometry is regarded as the gold standard for assessing muscle performance in both healthy individuals and clinical populations, supported by a multitude of studies that demonstrate the reliability and validity of the data obtained (e.g., [36]). Isokinetic dynamometry was utilized to evaluate the peak torque of concentric knee extension at a 60°/s angular velocity using the Kineo Training System (Kineo; V7, GLOBUS, Bolzano, Italy). The device was calibrated according to the manufacturer’s guidelines. During each testing session, M.S., a full professor with over 20 years of experience in strength testing and monitoring, conducted the isokinetic tests, while A.N. recorded all technical settings, such as belts and lever lengths. Subjects began the test seated, with their knees and hips flexed at a 90-degree angle. Upon the command of the strength and conditioning (S&C) coach, participants were strongly encouraged to exert maximal effort to fully extend their knees before gradually returning to the starting position. Each subject completed three trials, and the best performance from each velocity was chosen for further analysis. There was a 60 s rest period between legs. Participants performed three maximal repetitions for each leg. Lever length was used to calculate peak torque, with all force values weight-adjusted for both absolute and relative data (Nm·kg^–1^). The lever length was measured from the center of the knee joint to the lateral malleolus of the ankle joint. The formula for calculating peak torque was peak force × lever length × 9.81 m/s^2^.

### 2.4. Training Interventions

The groups adhered to the same weekly training schedule, as detailed in Table 1.

Participants from both groups engaged in a 6-week training program consisting of sessions held twice a week, with at least 48 h between strength training sessions. Strength training sessions were conducted during the initial phase of training, following a standardized warm-up and before the regular basketball practice. Strength training consisted of 2 sets of 8 repetitions of the half squat (0.010 kg/m^2^ moment inertia) or split squat (0.005 kg/m^2^ moment inertia) for weeks 1 to 3. For weeks 4 to 6, the training increased to 3 sets of 8 repetitions, adopting a suggested linear-periodization model with progressive increases in volume [37].

All sets were performed on a flywheel device (kBox; Exxentric AB, Bromma, Sweden), with a 2 min passive recovery between sets. To enhance momentum, an additional two repetitions were added at the beginning of each set, though these were not counted as part of the working set. For the half-squat exercise, participants began at a 90-degree knee angle and executed the concentric phase until full extension, before smoothly transitioning into the eccentric phase without stopping. Athletes were strongly encouraged to put forth maximal effort during the concentric phase and to postpone the braking phase until the final third of the downward movement, thereby creating eccentric overload.

Considering the split squat, players started in a lunge position with their hip and knee flexed at approximately 90 degrees. They accelerated the inertia load from the front leg until reaching near full extension of the hip and knee, then transitioned into the downward (eccentric) movement without pausing. Players were directed to complete the concentric phase as quickly as possible and with maximum effort, while postponing the braking phase until the final third of the downward movement, thereby creating eccentric overload.

### 2.5. Statistical Analysis

All tested variables were presented as mean ± standard deviation (SD). The normality of the data was verified using the Shapiro–Wilk test, while homogeneity was tested using Levene’s test. The differences in demographic data were analyzed using unpaired t-tests. Paired Student’s t-tests were utilized for within-group comparisons to identify significant differences between pre-test and post-test measurements in both study groups. Additionally, a two-way analysis of variance with repeated measures (training modality × time) was employed to examine the interaction and main effects. The statistical significance threshold was adopted at *p* ≤ 0.05. The effect sizes were calculated and classified according to Hopkins et al. [38], with the following thresholds: small (0.2), medium (0.5), large (0.8), and very large (1.3). All the data were analyzed using the SPSS package, version 25 (IBM Corp., Chicago, IL, USA).

## 3. Results

Demographic data between the UG (n = 11; age = 15.5 ± 0.5 years; stature = 186 ± 6 cm; body mass = 71 ± 12 kg) and BG (n = 11; age = 15.2 ± 0.4 years; stature = 183 ± 4 cm; body mass = 69 ± 7 kg) groups were compared. The results indicated no differences between groups in age, height, and body mass (*p* > 0.05)

There were no notable differences in the pre-test for any of the variables between the two groups. A significant interaction was identified only for the 5-0-5 tests (F = 13.27; *p* = 0.02; Table 2). In addition, significant pre–post improvements (8.4% and 13.3%) with large effect sizes (2.31 and 4.65) were demonstrated for the BG and UG, respectively.

No significant improvements from pre-test to post-test in both groups for EX60D (3.3%, *p* = 0.23 and 6.2%, *p* = 0.21 for the BG and UG, respectively) and EX60L (1.5%, *p* = 0.69 and 10.1%, *p* = 0.08, respectively) were reported. The UG group displayed greater effect sizes (0.5 and 0.33) compared to the BG group (0.09 and 0.19) for EX60L and EX60D, respectively.

When comparing the initial and final measurements for the CMJ test, the BG group demonstrated an 11.4% improvement (ES = 0.69, large effect size), while the UG group showed a 14.6% improvement (ES = 1.4, very large effect size). For the RSI, the BG and UG groups achieved significant improvements of 19.6%, with large effect sizes (0.95 and 0.77, respectively). In addition, while the BG group demonstrated significant pre–post improvements with large effect sizes in both unilateral jumps (13.9% and 11.9% for LCMJ and DCMJ, respectively), the UG demonstrated significant improvements for DCMJ (14.9%, large effect size) but not for LCMJ (7.0%, medium effect size). Moreover, for the SPR20m measure, notable differences were observed between pre-intervention and post-intervention assessments for both groups, although the percentage improvements were modest, at 2.3% for the BG and 2.8% for the UG, with effect sizes classified as trivial for all groups. In contrast, the comparison of pre- and post-intervention results for the SPR5m demonstrated significant enhancements in both cohorts, with improvements of 3.7% (small effect size) for the BG and 5.5% (large effect size) for the UG.

## 4. Discussion

Previous research demonstrated that both bilateral and unilateral strength training are effective in enhancing performance outcomes among athletes. However, there is a notable dearth of research examining the effectiveness of unilateral versus bilateral eccentric training specifically in the context of basketball. Consequently, the current study aimed to compare the impact of unilateral versus bilateral flywheel strength training on leg strength, jumping performance, change of direction, and sprinting ability in young basketball players. Our study results confirmed unilateral eccentric training to be superior to bilateral training in developing change of direction speed. Additionally, both groups exhibited significant improvements in change of direction, jumping, and sprinting performance, while strength levels demonstrated a nonsignificant upward trend, with the unilateral group showing greater enhancements. Based on the initial hypothesis, it appeared that incorporating two sessions per week of unilateral eccentric strength training with a low relative load for six weeks was a more effective approach for enhancing strength, jumping, sprinting, and change of direction performance compared to bilateral eccentric strength training in young basketball players.

Recent studies comparing unilateral and bilateral training effects on change of direction performance in athletes revealed inconsistent results. Appleby et al. [39] conducted an 18-week study with rugby players divided into unilateral, bilateral, and control groups. Both training modalities enhanced change of direction ability; however, bilateral training resulted in significantly greater improvements (effect size = 0.59 ± 0.64), potentially due to contraction specificity and the eccentric phase present in bilateral exercises. Drouzas et al. [40] and Stern et al. [41] found that preadolescent soccer athletes and youth players, respectively, showed significant improvements in agility tests across all groups, with no discernible differences between training modalities. Notably, Drouzas et al. found that agility scores improved by approximately 2%, whereas Stern et al. found performance changes of 0.31% and 0.61% for the bilateral group, and 1.52% and 2.58% for the unilateral group across the left and right legs, respectively. Our study reported larger increases of 8.4% and 13.3% for the bilateral and unilateral groups, respectively. Gonzalo-Skok et al. [42] observed similar patterns among highly trained young basketball players. Their findings indicated that unilateral training yielded greater enhancements in change of direction (2.6% improvement, ES = 0.48) compared to bilateral training (0.1%, ES = 0.02). An eight-week study on college male basketball players [19] indicated that unilateral training (Bulgarian split squats and reverse lunge jump squats) was superior to bilateral training (barbell rear squats and double-leg vertical jumps) in improving change of direction ability, with effect sizes demonstrating medium to large improvements (t-test: 1.584; 5-0-5 test left: 1.881). Collectively, these studies highlighted the varied impacts of unilateral and bilateral training on change of direction and agility performance. While some studies endorsed the superiority of unilateral methods, others suggested similar outcomes across both training modalities. These discrepancies may arise from differences in training protocols, sample characteristics, and performance metrics assessed.

Notably, there are only a few studies that have examined the impact of unilateral versus bilateral eccentric training on change of direction ability among athletes [29,30,31]. In a training design analogous to the present study, Núñez et al. [30] implemented a 6-week training program involving 27 young male team sports athletes, who were divided into a unilateral lunge group and a bilateral squat group. Each group engaged in two training sessions per week, performing four sets of seven repetitions of squats with a moment of inertia of 0.10 kg/m^2^ in the bilateral group and unilateral squats with each leg at a moment of inertia of 0.05 kg/m^2^ in the unilateral group, utilizing a flywheel inertial device (Exxentric kBox, Exxentric AB, Stockholm, Sweden). Pre- and post-intervention assessments of change of direction ability were conducted using four tests: 5 + 5-line sprints with 90° or 180° turns using both the dominant and non-dominant legs. The reported findings indicated that the unilateral group exhibited more substantial adaptations across all change of direction tests, as evidenced by greater mean effect sizes, compared to the bilateral group. The unilateral group demonstrated significant improvements in both changes of direction at 90°, with the magnitude of these changes (effect sizes ranging from 0.41 to 0.84 for the dominant and non-dominant leg, respectively) being lower than those observed in our study.

However, these changes are comparable to those recently reported for change of direction turns of 45° following either unilateral or bilateral training with an inertial device [29]. Moreover, the study showed significant interaction between groups in favor of the unilateral group for a specific change of direction parameter: the percentage of mean speed loss when executing change of direction maneuvers relative to the 10 m sprint time in the 90° non-dominant leg test (ES = 0.33). Overall, the unilateral group demonstrated more robust adaptations (i.e., higher mean effect size) in nearly all change of direction tests, corroborating both Núñez et al. [30] and our study findings that unilateral training leads to superior change of direction performance compared to bilateral training in athletes.

This phenomenon can be elucidated through the characteristics of change of direction movements, which often necessitate unilateral force generation and support from the legs. Unilateral training more effectively simulates the movement patterns associated with change of direction performance, thereby directly enhancing the relevant muscle groups and neuromuscular coordination [41]. Furthermore, Mausehund et al. [43] conducted a comparative analysis of muscle activation during barbell lunge, step-ups, and rear elevated split-squat exercises, revealing that the split squat elicited the highest level of hamstring activation. Consequently, when applied with equivalent relative loads, the split-leg squat was more effective in developing hamstring strength and facilitating the improvement of technical movements that significantly involved the hamstrings, such as change of direction.

No notable effects were observed of either unilateral or bilateral training on strength among our participants. Previously, flywheel strength training has been shown to be effective in improving muscle strength in short-term strength training periods [44], with particularly strong evidence (overall effect size of 1.33) in trained young individuals [45]. These findings are inconsistent with those of our study, which utilized a flywheel device over a six-week period, training twice per week with low moment inertia settings of 0.010 kg/m^2^ for the BG and 0.005 kg/m^2^ for the UG. This regimen resulted in nonsignificant increases in muscle strength, with increases of 1.5% and 3.3% in the BG and 10.1% and 6.2% in the UG for the left and right leg, respectively.

It is worth noting the clear trend of greater improvements in the UG. While the improvements observed in the unilateral group were not statistically significant, their magnitude aligned with some, but not all, existing literature. For instance, one study reported an increase of 8.6% (effect size (ES) = 0.59) in one-repetition maximum (1RM) back squat performance among female amateur basketball athletes following a flywheel strength training protocol consisting of four sets of ten repetitions, conducted twice per week over a four-week period [26]. Conversely, another study documented an increase of 18.7% (ES = 1.883) in isometric half-squat strength among junior basketball players following a flywheel strength training protocol consisting of two to four sets of eight repetitions, conducted once/twice per week over an eight-week period [25].

It can be posited that the observed nonsignificant improvement in strength may be influenced by the divergence in training and testing modalities (closed versus open kinetic chain). Previous research indicated a moderate correlation between closed and open kinetic chain assessments of muscular strength (*r* = 0.64) [46]. Therefore, it is reasonable to assume that isokinetic testing may lack the specificity required to effectively evaluate the distinct functional adaptations achieved through training modalities, such as the half squat for bilateral groups and the single-leg squat for unilateral groups.

The RSI measures a person’s ability to swiftly transition from an eccentric muscle contraction to a concentric one [47]. It is reasonable to postulate that flywheel training, designed to enhance eccentric force production within a shorter timeframe, will result in improvements in the RSI. Furthermore, the RSI has demonstrated a strong correlation with the change of direction speed, acceleration speed [48], and agility [49], important determinants of basketball performance.

To the extent of our knowledge, our studies constitute the inaugural investigation into the impact of flywheel strength training on the RSI among adolescent basketball players. The study revealed substantial, identical pre-to-post improvements of 19.6%, accompanied by large effect sizes of 0.95 and 0.77 for the bilateral jumping (BG) and unilateral jumping (UG) assessments, respectively. Although there is a scarcity of research focusing on the effects of flywheel training on RSI, the observed alterations in our study surpass those documented in prior investigations within a comparable demographic. In the study by Fiorilli and colleagues [50], the RSI was not improved after six weeks of flywheel training (pre-test: 1.94 ± 0.40; post-test: 2.07 ± 0.42) in youth soccer players. Similarly, Murton et al. [51] reported no significant improvements in the RSI after four weeks of twice per week flywheel training in elite male academy rugby players (pre-test: 0.84 ± 0.18; post-test: 0.83 ± 0.18).

The obtained differences of our study results compared to the aforementioned studies could be attributed to flywheel training vector specificity and training load, respectively. Indeed, the vertical (half squat and split squat) movements in our study compared to horizontal (running)-oriented movements applied in Fiorilli et al.’s study have been proven to produce greater training effects on vertical jump performance [29]. In addition, the six-week training program with the load of 0.010 kg/m^2^ (bilateral) or 0.005 kg/m^2^ (our study design) represents an overall higher load in comparison to four weeks with the load of 0.005 kg/m^2^ (bilateral) reported by Murton et al. [51]. It is generally recognized that higher strength training loads produce greater strength/power adaptations [8] and, consequently, enhance sport-specific tasks, such as various jumping parameters [52].

The CMJ performance, both bilateral (11.4%, ES = 0.69 and 14.6%, ES = 1.4 for the BG and UG, respectively) and unilateral (left leg: 13.9% (ES = 0.6) and 7% (ES = 0.39); right leg: 11.9% (ES = 0.52) and 14.9% (ES = 0.61), for the BG and UG, respectively) showed significant pre–post increases, but without an inter-group effect. This percentage of change and the change’s magnitude are higher than those recently reported by Núñez et al. [30], Gonzalo-Skok et al. [29], and Hernández-Davó et al. [31].

The superior improvements in our study compared to the aforementioned studies could be attributed to the following:Dynamics of effort—Hernández-Davó et al. [31] and Gonzalo-Skok et al. [29] employed higher loading conditions of 0.025 kg/m^2^ and 0.027 kg/m^2^, respectively. These increased loading conditions likely resulted in diminished execution velocities, which in turn could have led to a reduced transfer of training effects on vertical jump performance [53].Study subjects’ age—It has been reported that adolescents, a demographic represented in our study sample, show distinct muscular adaptations by exhibiting higher gains in explosive power in response to resistance training compared to adults, as represented in the Núñez et al. [30] and Pesta et al. [54] studies.

Furthermore, unilateral training (UG) demonstrated a more pronounced impact on jump performance compared to bilateral training. This observation contradicts the findings of Núñez et al. [30] and Gonzalo-Skok et al. [29] yet aligns with the results reported by Hernández-Davó et al. [31]. Notably, unilateral eccentric training seemed to elicit significant bilateral adaptations during bilateral jumping activities, suggesting that the cumulative strength developed in each leg, when trained independently, may enhance overall bilateral performance.

Both training modalities induced significant enhancements in linear straight sprinting, with a greater magnitude in shorter distances (3.7%, ES = 0.7 and 5.5%, ES = 1.08 for the BG and UG, respectively) than in longer distances (2.3%, ES = 0.53 and 2.8%, ES = 0.77 for the BG and UG, respectively). Collectively, these results are in accordance with some, but not all, previous results found following flywheel programs in similar cohorts.

O’Brien et al. [26] reported positive effects on the 10 m sprint time (2%, ES = 0.54) after four weeks of two sessions per week of flywheel training in female basketball players. A similar study [55] reported a 2.3% and ES = 0.67 decrease in their 20 m sprint time following a seven-week flywheel training program. The largest improvements in sprinting ability after a flywheel strength training program were reported by Stojanović et al. [25] in junior basketball players. Following an eight-week flywheel strength program with two to four sets of eight repetitions, significant 10.3% (ES = 3.79) improvements were presented for 5 m sprints. Contrary to the aforementioned findings, Raya-Gonzales et al. [56] reported no significant changes in sprinting performance (10 m, 20 m, and 30 m) in young soccer players after the completion of a 10-week (one session per week, 2–4 sets of 8–10 reps of the lateral squat with an inertia of 0.025 kg·m^2^) flywheel strength training program. Moreover, Coratella et al. [57] found no significant impact on 10 m and 30 m sprint performance after an eight-week squat flywheel training regimen. The study involved 40 male soccer players who participated in a weekly session over the course of 8 weeks, which included 48 squat repetitions using a flywheel device (inertia = 0.11 kg·m^2^). The results showed no meaningful improvements in 10 m or 30 m sprint times following the training, with changes of only 2% (ES = 0.33) for the 10 m and 2% (ES = 0.32) for the 30 m sprints. The authors concluded that a single session each week may not provide sufficient stimulus to improve linear sprinting performance. Collectively, the existing literature presents inconsistent evidence regarding improvements in sprint performance attributable to flywheel training, highlighting the necessity for further investigations to ascertain its efficacy.

Our study has its limitations. Firstly, since our data were gathered from a small sample size, the findings can only be generalized to comparable samples of subjects and competition levels. Furthermore, we did not track the load during the flywheel strength sessions, which could have offered us further insights into the overall load applied and the potential dose/response training effects. Lastly, our study was limited to just six weeks, so longer comparative investigations involving these specific strength training modalities are necessary.

## 5. Conclusions

It appeared that incorporating two sessions per week of unilateral flywheel strength training with a low relative load for six weeks was a more effective strategy for enhancing strength, jumping, sprinting, and change of direction performance compared to bilateral flywheel strength training in young male basketball players. Strength and conditioning professionals should consider adopting unilateral flywheel training during the season to achieve significant improvements in these performance metrics.

## Figures and Tables

**Table 1 jfmk-10-00081-t001:** Weekly training schedule.

Monday	Tuesday	Wednesday	Thursday	Friday	Saturday	Sunday
**Off**	Flywheel strength training + Team practice	Team practice	Flywheel strength training + Team practice	Team practice	Match	Recovery

**Table 2 jfmk-10-00081-t002:** Pre–post and between-group differences in selected variables with % of improvement and Cohen’s effect size (d).

	BG				UG			
	PRE	POST	%	d	PRE	POST	%	d
5-0-5 test (s)	2.50 ± 0.10	2.29 ± 0.08 *	8.4	2.31	2.63 ± 0.08	2.28 ± 0.07 *†	13.3	4.65
EX60L (Nm)	198.16 ± 38.96	201.16 ± 24.24	1.5	0.09	190.59 ± 29.12	209.91 ± 43.56	10.1	0.5
EX60D (Nm)	198.13 ± 38.90	204.66 ± 26.39	3.3	0.19	193.27 ± 32.77	205.28 ± 37.73	6.2	0.33
RSI (s)	1.79 ± 0.33	2.14 ± 0.40 *	19.6	0.95	1.63 ± 0.41	1.95 ± 0.42 *	19.6	0.77
CMJ (cm)	33.46 ± 5.92	37.27 ± 5.00 *	11.4	0.69	33.40 ± 3.36	38.27 ± 3.55 *	14.6	1.4
LCMJ (cm)	15.55 ± 3.67	17.71 ± 3.52 *	13.9	0.6	17.52 ± 3.60	18.75 ± 2.48	7.0	0.39
DCMJ (cm)	17.14 ± 3.68	19.18 ± 4.14 *	11.9	0.52	15.69 ± 3.88	18.02 ± 3.69 *	14.9	0.61
SPR5m (s)	1.07 ± 0.07	1.03 ± 0.06 *	3.7	0.7	1.10 ± 0.06	1.04 ± 0.05 *	5.5	1.08
SPR20m (s)	3.17 ± 0.14	3.10 ± 0.12 *	2.3	0.53	3.19 ± 0.10	3.10 ± 0.13 *	2.8	0.77

* 5-0-5 test—change of direction test; EX60L—left leg knee extension test (60°/s); EX60D—right leg knee extension test (60°/s); RSI—reactive strength index test; CMJ—countermovement jump test; LCMJ—left leg countermovement jump test; DCMJ—right leg countermovement jump test; SPR5m—5 m sprint test; SPR20m—20 m sprint test; IN—initial test result ± standard deviation; FIN—final test result ± standard deviation; %—percentage of improvement; * statistically significant difference pre vs. post, *p* < 0.05; †—significantly different BG vs. UG.

## Data Availability

Data are available from the author upon reasonable request.

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
