# Peer review of "A Comparison of Bilateral vs. Unilateral Flywheel Strength Training on Physical Performance in Youth Male Basketball Players"

_jfmk, 2025, doi:10.3390/jfmk10010081_

Round 1
Reviewer 1 Report
Comments and Suggestions for Authors
Thank you very much for doing this research work.
I congratulate you on your work, but would like to make some suggestions for improvement.
- Keywords: unilateral flywheel training; bilateral flywheel training (better to modify these two terms). It is also recommended to use more concrete terms and not to use several words.
- Line 72: say what the acronyms CMJ and COD stand for.
- Material and methods: authors should be asked to specify the protocol. They do describe what they do, but if you want to reproduce the protocol, it is not described in a way that can be reproduced (study design section).
- Measurement test: no bibliographical references are shown. It is requested that this be reviewed and modified to reference the information provided.
- The results section is too short compared to the discussion (too long).
- The socio-demographic data of the sample should be shown in the results section.
- Too little information is provided in the table (results). A more detailed analysis of the results should be done.
- Discussion: lines 273 to 287 can be framed within the conclusion of this study. - The discussion section presents the results of other studies in comparison with this one, but I consider that it should be more concise and group studies that show common characteristics in order to be able to specify and reduce the length of this section, discarding extra information, which, although they could answer questions related to this study, should not be so developed as it would mean extra work for the readers to understand.
- Line 288: reference what it indicates (numerous studies ....).
- Line 344: hyphenate references.
- References: check those references that are older than 10 years, as well as a revision in the statement (some citations do include the authors' doi, while in others it does not appear). If it is not possible to update, explain in the text the reason for the update.
Reviewer 2 Report
Comments and Suggestions for Authors
Please clarify the sex of the participants.
L11. “impacts”. Maybe better to replace with “effects”.
Please ensure that abbrevations are defined on first use, e.g. EX60 in the abstract.
Please ensure throughout the manuscript that there is no dot before the reference, e.g. change “.[1].” To “[1].
L38. Please delete “etc.” or elaborate.
L64. Ref 22 is on soccer.
Ls 85-88. I suggest to express age values with one decimal place and stature and body mass without decimal places.
L86. There are n=11 in both groups. How was the randomization? Was it allocation? Please clarify.
L89. Two different clubs, so members of one club did one type of training? Please clarify.
L90. Please provide name of the country.
L91. How many levels is the regional league below the national league. I suggest also to consult McKay et al 2022 (i.e. doi: 10.1123/ijspp.2021-0451) for categorisiation of the athletes.
L102. Was this 90% for the flywheel training sessions. Please clarify.
Ls 145-147. Was this precision needed? These were not primary outcomes?
Ls 149 and 156. Please ensure consistency in description of the force plate.
L193. The mention of an individual with experience seems to suggest that other tests were executed with guidance by individuals with less experience.
L235. An analysis of the differences between pre- and post-measurements requires an unpaired t-test, as you have two parallel groups. In addition, the two-way ANOVA could be used for analysis of time of the absolute values. Why was that not done. The authors need to reconsider the statistical analysis of the data.
L253. Change “(60 /s)” so that it has a degree sign.
Table 2. I suggest to replace IN and FIN with pre and post.
Replace the “,” with a “.” In percentage values, e.g. 8,4% to 8.4%.
L247. “nonsignificant trends” needs justification/clarification. What were the p-values?
Round 2
Reviewer 2 Report
Comments and Suggestions for Authors
The authors have not consistently responded to some comments, so these will appear here again.
Please clarify the sex of the participants in the title and the conclusions.
Throughout the manuscript, please express mean and SD age values with one decimal place and stature and body mass without decimal places.
Please ensure that abbrevations are defined on first use, e.g. EX60 in the abstract. 60 is for 60 degrees/s and that is not clear as of now.
L205. Plural for studies but only one reference. Please change “[36]” to “[e.g. 36].
L213. Please change “exerte” to “exert”
L253. An analysis of the differences between pre- and post-measurements requires an unpaired t-test, as you have two parallel groups.
Ls 260-263. Please ensure consistency in font type.
Throughout the manuscript, please replace the “,” with a “.” In percentage values, e.g. 8,4% to 8.4%. But also for number, e.g. L266 change “4,65” to “4.65”. Please be meticulous in your response.
L268. “nonsignificant trends”. Maybe only the p-value of 0.08 justifies the mention of a non-significant. The others observations are not improvements as the p-values indicate no change. Please revise and recognize that p-values >0.1 do not allow statements of improvements.
L339. Nunes is incorrect. Please revise throughout the manuscript.
Author Response
please find responses to reviewer comments in attach

Round 3
Reviewer 2 Report
Comments and Suggestions for Authors
Please express mean and SD of age values with one decimal place and stature and body mass without decimal places.
i.e. Ls Demographic data between UG (n=11; age=15.45±0.52 years; stature=186.10±5.99 cm; 262
body mass=71.38±12.08 kg) and BG (n=11; age=15.18±0.41 years; stature=183.34±3.99 cm; 263
body mass=69.14±7.47 kg) groups were compared.
Regarding the analysis, you state that you analyse the differences, i.e. post-pre for unilateral training vs post-pre for for bilateral. So it seems you created two columns of data (with differences) with each column from a different training group. That analysis requires a unpaired t-test.
Your demographic data, i.e Ls 262-264 also requires a unpaired t-test.
I hope this clarifies, otherwise please seek statistical assistance with the dataset within your institute.
Author Response
please find attach.
